# No Evidence of Neutrophil Response Modulation in Goats after Immunization against Paratuberculosis with a Heat-Inactivated Vaccine

**DOI:** 10.3390/ani14111694

**Published:** 2024-06-05

**Authors:** Miguel Criado, Marta Silva, Pedro Mendívil, Elena Molina, Valentín Pérez, Julio Benavides, Natalia Elguezabal, Daniel Gutiérrez-Expósito

**Affiliations:** 1Departamento de Sanidad Animal, Instituto de Ganadería de Montaña (IGM) CSIC-ULE, Ctra León-Vega de Infanzones, 24346 León, Spain; msilvr@unileon.es (M.S.); pmeng@unileon.es (P.M.); vperp@unileon.es (V.P.); julio.benavides@csic.es (J.B.); dgute@unileon.es (D.G.-E.); 2Departamento de Sanidad Animal, Facultad de Veterinaria, Universidad de León, Campus de Vegazana s/n, 24071 León, Spain; 3Departamento de Sanidad Animal, NEIKER-BRTA, Instituto Vasco de Investigación y Desarrollo Agrario, 48160 Derio, Spain; emolina@neiker.eus (E.M.); nelguezabal@neiker.eus (N.E.)

**Keywords:** neutrophils, NETs, vaccination, mycobacterium, paratuberculosis, cell-mediated immunity, cytokines, ruminants

## Abstract

**Simple Summary:**

Among other limitations, the currently available vaccines against paratuberculosis do not offer complete protection against infection, and further vaccine development is limited by a lack of understanding of the mechanisms behind vaccine-induced protection. In this regard, the most recent studies have demonstrated that neutrophil function can be modulated through vaccination against several pathogens, including *Mycobacterium avium* subspecies *paratuberculosis* (*Map*). However, this modulation has not been described in ruminants, which are the natural hosts of *Map*. In the present work, the effect of vaccination on the neutrophil response against *Map* was assessed in goats using the only available vaccine against small ruminant paratuberculosis, Gudair^®^. No differences were found in the ex vivo response of neutrophils isolated from non-vaccinated and vaccinated animals, which suggests that the protection conferred by this heat-inactivated vaccine is based on mechanisms other than neutrophil modulation. It is possible that neutrophil modulation depends largely on the intensity of the immune response elicited by the vaccine employed or the antigen dose, as the previous reports which observed this modulation used live attenuated vaccines or were performed in laboratory animals using experimental vaccines.

**Abstract:**

Neutrophils are believed to play a role in the initial stages of paratuberculosis, and it has recently been demonstrated that vaccination can modulate their function via priming or through epigenetic and metabolic reprogramming (training). Modulation of the neutrophil response against *Mycobacterium avium* subspecies *paratuberculosis* (*Map*) through vaccination has been demonstrated in a rabbit model but not in ruminants. Therefore, in the present work, the effect of vaccination on the response of caprine neutrophils against *Map* was studied. Neutrophils were isolated from non-vaccinated (*n* = 7) and Gudair^®^-vaccinated goat kids (*n* = 7), before vaccination and 30 days post-vaccination. Then, several neutrophil functions were quantified ex vivo: cell-free and anchored neutrophil extracellular trap (NET) release, phagocytosis, and the differential expression of several cytokines and TLR2. The induction of cell-free NETosis and TLR2 expression by *Map* is reported for the first time. However, vaccination showed no significant effect on any of the functions studied. This suggests that the protection conferred by Gudair^®^ vaccination is based on mechanisms that are independent of the neutrophil function modulation. Further research into the impact of alternative vaccination strategies or the paratuberculosis infection stage on ruminant neutrophil function could provide valuable insights into its role in paratuberculosis.

## 1. Introduction

Ruminant paratuberculosis (PTB), caused by the *Mycobacterium avium* subspecies *paratuberculosis* (*Map*), is a widespread mycobacterial infection that is responsible for significant economic losses in the livestock industry [1]. Vaccination is considered to be the most effective control measure [2,3]; however, the only commercially available heat-inactivated vaccines, such as Gudair^®^ and Silirum^®^, share a number of drawbacks: (i) they induce the formation of an injection site granuloma; (ii) they interfere with tuberculosis and PTB diagnosis; and (iii) they do not confer complete protection against infection [4,5]. Effective vaccine development is hindered by a lack of knowledge regarding the mechanisms implicated in PTB pathogenesis and, in particular, the host immune response against both infection and vaccination. In fact, a high individual variability has been observed between vaccinated animals after infection, and this needs to be clarified [2]. In this regard, the cross-talk between *Map* and its most important target cell, the macrophage, has been studied in depth in infected and vaccinated animals, using various in vivo and ex vivo [6,7,8,9,10] approaches. However, evidence related to the role of neutrophils in the immune response against *Map* infection has been accumulating in recent years. For instance, it is well known that they are present in the initial phases of the infection; however, their numbers decrease as the disease advances [11,12,13], and recently described changes in the local transcriptome of infected animals suggest that an impairment of neutrophil recruitment and activation occurs in the later stages of the disease [14,15].

Over the last two decades, new in vitro culture techniques have allowed a deeper understanding of these cells; however, studies on ruminants are still scarce [16,17,18]. A new effector function, the formation of neutrophil extracellular traps (NETs), was first described in 2004 [19], and a regulatory role of neutrophils (the expression and production of both pro- and anti-inflammatory cytokines during the early stages of infection) is being revealed [20,21,22]. In this regard, the advancements in cell function assays have made it possible to demonstrate that neutrophils employ their “classic” effector functions against *Map* and, above all, that they efficiently kill this bacterium [16,18,23]. NETosis against *Map* has recently been described in vitro in several ruminant species: cattle [16], sheep [24], and goats [23]. In vivo induction of NETs by *M. tuberculosis* (*Mtb*) has already been proven in a guinea pig model [25], and even though its presence in PTB infection in vivo is still unknown, an extracellular *Map* DNase (MAP3916c) which destroys NETs in vitro has been shown to be necessary for *Map* virulence, suggesting that this mechanism occurs in vivo as well [26]. With regard to the modulating role of neutrophils, an increase in proinflammatory cytokine expression (IL-1β and TNF) [23] and production (IL-1β and IL8) [16] by caprine and bovine neutrophils, respectively, has been observed after *Map* exposure ex vivo.

In addition to the neutrophil’s effector functions and its ability to modulate the immune response through cytokine production, a new concept, “trained immunity”, based on the long-term epigenetic and metabolic reprogramming of the innate immune cells, is gaining attention [27,28]. Functionally, trained immunity works as an innate immune memory which provides protection against both homologous and heterologous insults [29]. The immunological phenotype of trained immunity has been proven to last at least 3 months and up to 1 year [30]. In neutrophils, “training” or innate memory has generally been studied between 7 days and 3 months after the primary stimulus [18,31,32,33]. The human tuberculosis vaccines Bacillus Calmette–Guérin (BCG) and MTBVAC, which are based, respectively, on attenuated strains of *Mycobacterium bovis* (*Mbv*) and *Mtb*, are well-known inducers of the neutrophil’s trained immunity [34]. In this regard, it has been demonstrated that BCG vaccination generates a long-term functional reprogramming of these cells, increasing their antimicrobial functions [32]. Also, the induction of specific Th1 and Th17 responses as a consequence of neutrophil modulation by BCG, has demonstrated in vivo bacterial load reduction [35].

In goats vaccinated against PTB, partial cross-protection has been observed when challenged with *Mbv* [36]. In cattle, it has been found that vaccination against PTB reduces mortality and losses that are not attributable to this disease [37], and heterologous protection against *Mbv* has also been observed [38]. This parallels findings from previous studies on human BCG vaccination which demonstrated similar effects [39,40,41] and suggest a possible mechanism of trained immunity and/or cross-protection in ruminants. The molecular basis of neutrophil trained immunity is still being investigated [32]. In PTB, neutrophil response modulation through vaccination has only been investigated through functional studies. In this regard, immunization with Silirum^®^ or an attenuated oral *Map* vaccine, and even with an inactivated *Corynebacterium pseudotuberculosis* (*Cpstb*) oral vaccine, in a rabbit model increased *Map* phagocytosis by neutrophils and induced an increase in NETosis against *Map* [18]. In human BCG vaccination studies, neutrophil phagocytosis was increased, but not NET release [32]. In sheep, Silirum^®^ vaccination did not affect NETosis in response to *Map* or other pathogens such as *Staphylococcus aureus* and *Escherichia coli* [24]. Though contradictory, some of these findings suggest that neutrophils could participate in the trained immune response against *Map*; however, they also show that there are numerous knowledge gaps in this process. Moreover, most of the studies demonstrating the potential modulation of neutrophil function through vaccination have been conducted using experimental models that differ from the target species of the assessed vaccine [18,31,33,35,42]. In this regard, the effect of vaccination against *Map* on neutrophil function has not been explored in ruminants such as goats or cattle, and, as previously stated, no influence of vaccination on NETosis against *Map* has been observed in sheep [24].

Therefore, the main objective of the present study was to assess the effect of vaccination with Gudair^®^ (authorized for use in small ruminants) on goat neutrophil functionality after ex vivo *Map* infection. To achieve this, goat kids were vaccinated in the same manner as they would be in field conditions; afterwards, a standardized neutrophil isolation protocol and a series of techniques, including immunofluorescence, fluorometry, flow cytometry (FCM), qPCR, and RT-qPCR, were applied to quantify NETosis, phagocytosis, and cytokine and TLR2 expression by the neutrophils.

## 2. Materials and Methods

### 2.1. Ethics Statement

The goat handling and blood sample collection were conducted in compliance with EU legislation (Law 6/2013) regarding exploitation, transportation, and experimentation; R.D. 118/2021 for the protection of animals used in research and teaching; and Directive 2010/63/EU on the protection of animals used for scientific purposes. All the procedures were approved by the corresponding animal welfare body (OEBA) and the Consejería de Agricultura y Ganadería de la Junta de Castilla y León (authorization code ULE-02-2021). All the animals used in this study were handled in strict accordance with good clinical practices, with all efforts made to minimize suffering.

### 2.2. Animals and Experimental Design

Fourteen healthy one-month-old, mixed-breed female goat kids, randomly selected and acquired from a commercial flock without a previous history of PTB, were used in this study. They tested negative for the PTB antibody ELISA (ID Screen^®^ Paratuberculosis Indirect, IDvet, Gabrels, France) and for the interferon-γ release assay (BOVIGAM™ TB Kit, Thermo Fisher Scientific, Waltham, MA, USA) [23]. The goat kids were housed together in the experimental facilities of the Instituto de Ganadería de Montaña (IGM, CSIC-ULE) and followed a diet based on grass hay *ad libitum* and a conventional compound feed throughout the entire experiment.

Animals were randomly divided into non-vaccinated (*n* = 7) and vaccinated (*n* = 7) groups. The latter group was vaccinated with Gudair^®^ (CZ Vaccines, Porriño, Spain). This is the only currently approved vaccine for use against PTB in small ruminants, and we aimed to replicate the actual conditions of its use. Therefore, it was administered according to the manufacturer’s recommendations: a single, 1 mL subcutaneous dose was used to inoculate one-month-old female goat kids. The recommended age for vaccination is between 1 and 6 months, and earlier timepoints seem to confer better protection [43]. On the other hand, the non-vaccinated group was inoculated with 1 mL of phosphate buffered saline (PBS) via the same route. 

All the animals were sampled 1 day before vaccination (day 0) and 30 days post-vaccination (dpv) (day 30). In each sample, blood was collected from the jugular vein into heparinized Vacutainer^®^ tubes, to isolate the neutrophils and perform multiple ex vivo assays, as explained below. Moreover, by 30 dpv, the presence of specific antibodies against *Map* was confirmed in all the vaccinated animals using the same ELISA test mentioned above (ID Screen^®^ Paratuberculosis Indirect, IDvet, Gabrels, France), whereas the non-vaccinated animals remained seronegative (Appendix A). The ELISA was performed according to the manufacturer’s instructions. Briefly, the serum samples were incubated with a neutralization buffer, which included inactivated *Mycobacterium phlei*. After neutralization, the samples were incubated in 96-well plates coated with *Map* antigen. After the washing steps, the samples were incubated with an anti-ruminant IgG-peroxidase conjugated antibody. Finally, after the washing steps, the chromogenic substrate was added, and the reaction was stopped with H_2_SO_4_ 0.5 M after 15 min. The absorbance values were measured spectrophotometrically at 450 nm using an ELX800 ELISA reader (Bio-Tek Instruments, Winooski, VT, USA). The S/P was calculated by dividing the corrected (via the subtraction of the negative control absorbance) sample absorbance by the corrected positive control absorbance and was expressed as a percentage (S/P %).

### 2.3. Neutrophil Isolation

The neutrophils were isolated from heparinized blood using a previously described method [23]. Briefly, the blood was diluted in PBS and centrifuged, and the buffy coat was collected, resuspended in PBS, layered on top of an equal volume of Lymphoprep^®^, and centrifuged again. Then, the granulocyte layer was transferred to another tube, and after erythrocyte lysis and washing, the neutrophils were resuspended in the incubation media, which was used for all the assays: Gibco™ RPMI 1640 (11835-063, Thermo Fisher Scientific, Waltham, MA, USA) supplemented with 2% heat-inactivated fetal bovine serum (10500064, Gibco^®^, Paisley, UK) and L-glutamine (25030081, Gibco^®^, Paisley, UK) at a final concentration of 2 mM. The high viability (>98%) and purity (>90% of the neutrophils, with the remaining cells being eosinophils) of the final isolation product were determined as previously described [23]. 

The neutrophils were then counted on a cell counter (Corning, Corning, NY, USA), and the concentration was adjusted depending on the assay, as further explained below. Then, the neutrophils were seeded in different culture plates to perform the NETosis quantification and gene expression assays, or in tubes for the phagocytosis assay, as explained in the following sections.

### 2.4. Bacteria Preparation

The *Map* K10 strains (with and without a green fluorescent protein (GFP)-expressing plasmid) were grown to an exponential phase, aliquoted after concentration adjustment, and frozen, as previously described [23]. 

Before the assays were performed, the *Map* aliquots were thawed and resuspended in fresh medium (7H9-OADC-MJ for *Map* K10 or 7H9-OADC-MJ-Kan for *Map* K10-GFP) and incubated for 3 h at 37 °C. Afterwards, the bacteria were washed twice and resuspended in neutrophil culture medium, passed through a 27-gauge syringe needle, and vigorously vortexed to disperse clumps before in vitro infection. An MOI of 10 was used in all the assays; this was based on our previous work, in which a significant response of caprine neutrophils against *Map*—without deleterious effects—was observed using some of the techniques applied in this study [23]. This MOI has also been used in previous works assessing the effect of vaccination on neutrophil response against *Map* [18,24].

### 2.5. Immunofluorescence

Neutrophil culture, immunofluorescence staining, and imaging were performed as previously described [23]. Briefly, a total of 5 × 10^5^ neutrophils were seeded in duplicates on sterile poly-l-lysine pre-coated cover glasses in 24-well plates. They were incubated for 3 h with *Map*-GFP, using an MOI of 10, 50 nM PMA (positive controls), or left unstimulated (negative controls). Afterwards, the cells were fixed, permeabilized, and blocked. The slides were washed and immunofluorescence staining was performed overnight using the rabbit anti-MPO Alexa Fluor 750 conjugated antibody (BS-4943R-A750, Bioss, Woburn, MA, USA) at a 1:200 dilution and the mouse anti-pan-histone primary antibody (MAB3422, Merck, Darmstadt, Germany) at a 1:400 dilution, coupled with 1 h incubation using the secondary antibody goat anti-mouse IgG (H + L) cross-adsorbed AF647 (Invitrogen, Carlsbad, CA, USA). Finally, the coverslips were mounted on slides using Fluoroshield™ with DAPI (4′,6-diamidino-2-fenilindol) (Sigma-Aldrich, St. Louis, MO, USA). Imaging was performed using the direct microscope Eclipse Ni-E (Nikon, Tokyo, Japan) equipped with the Prime BSI Scientific CMOS scientific camera (Photometrics^®^ Prime BSI™, Scottsdale, AZ, USA).

### 2.6. NETosis Quantification

Two different assays were used: the first was based on the quantification of NETs through immunofluorescence microscopy using the colocalization of DNA and histones; the second method was based on the quantification of DNA release through fluorometry [23], with a few modifications taken from Tanaka et al. [44]. These techniques complement each other given that the fluorometric quantification of DNA is very sensitive to small changes, but they do not discriminate between NETosis and the DNA released from dead neutrophils [23].

First, the coverslips used for NETosis visualization (see Section 2.5) were used to perform an estimation of NETosis. Briefly, for each animal and treatment (untreated, 50 nM PMA and *Map* at an MOI of 10), two slides were examined with an Eclipse Ni-E (Nikon, Tokyo, Japan) microscope, equipped with the DS-Ri2 color microscope camera (Nikon, Tokyo, Japan), and 10 random fields (400×) with equal neutrophil densities (149.4 neutrophils/field on average) were analyzed. A field was considered as positive if at least one neutrophil was clearly undergoing NETosis, as identified by the release of both DNA (labelled with DAPI) and histones (labelled with the anti-pan-histone antibody) in a branching pattern. This approach was chosen to address the challenge of distinguishing exactly which specific cells were undergoing NETosis, as the neutrophil aggregates induced by mycobacterial clumps made it impossible to quantify which cells were releasing NETs. The data were expressed as an increase in fields with NETs and were obtained by subtracting the percentage of positive fields in the control samples from the treated samples.

Second, to quantify the DNA release for each animal and treatment (untreated, PMA and *Map*), 2 × 10^5^ neutrophils were seeded in triplicate in a 96-well plate and incubated for 3 h, as previously described [23]. Then, to quantify the extracellular, cell-free NETs, which are released away from the neutrophils [44], the culture supernatant of each well was collected and transferred to another well, and the dsDNA in the culture supernatants was measured using a PicoGreen dsDNA assay kit. After removing the supernatants, this kit was also used to quantify the anchored NETs in the cells left at the bottom of the wells, using the same method mentioned above [44]. Briefly, the samples were excited at 480 nm, and the fluorescence emission intensity was measured at 520 nm using the BioTek Synergy H1 multimode microplate reader; the extracellular DNA concentration was calculated using the PicoGreen standard curve, as previously described [23]. Then, for each animal, the increase in anchored and cell-free NETosis, induced by PMA and *Map,* with respect to the untreated neutrophils, was calculated using the mean DNA concentration of each sample and expressed as a percentage. 

### 2.7. Quantification of Neutrophil Map Phagocytosis Using Flow Cytometry

Neutrophil phagocytosis was measured using FCM, as previously described [23]. Briefly, for each animal, 2.5 × 10^6^ neutrophils with and without *Map*-GFP at a MOI of 10 were incubated in suspension for 20 min at 37 °C, in a tube rotator at 6 rpm and a 75-degree angle. Non-incubated and non-labelled controls were used to ensure neutrophil integrity and to adjust the fluorescence threshold. Afterwards, the samples were constantly kept at 4 °C, fixed (2% paraformaldehyde for 10 min), permeabilized (70% ethanol for 10 min), and resuspended in PBS with 1% bovine serum albumin (BSA). The neutrophils were then incubated for 30 min with the rabbit anti-MPO polyclonal antibody AF750 at a 1:50 dilution, washed twice, and resuspended in PBS with 1% BSA. The FCM data were acquired using the Cytek Aurora^®^ flow cytometer (Cytek Biosciences, CA, USA) and processed using SpectroFlo^®^ software version 3.1.0 (Cytek Biosciences, Fremont, CA, USA). The gating strategy is detailed in Appendix A.

For each animal, the percentage of GFP+ neutrophils after incubation with *Map*-GFP was calculated by establishing a threshold (as seen in Appendix A) using the maximum fluorescence intensity (FI) of the neutrophils incubated without bacteria, as previously described [18,23]. In addition, for each animal, the mean increase in GFP intensity was calculated by dividing the mean GFP FI of the neutrophils incubated with *Map*-GFP by the mean GFP FI of the neutrophils incubated without bacteria and was expressed as a percentage.

### 2.8. Map DNA Quantification through qPCR 

For each animal, a total of 5 × 10^5^ neutrophils were seeded and cultured with and without *Map* at a MOI of 10, in duplicate, in 24-well culture plates and incubated for 3 h at 37 °C in a 5% CO_2_ atmosphere. After incubation, the supernatants were collected, and the cells were washed with warm PBS to collect the remaining non-phagocyted or lightly adhered bacteria; the volume of this wash was added to the supernatant sample [16]. Afterwards, the neutrophils were scraped in cold PBS using an inverted pipette tip; after the cells were collected, the wells were washed with cold PBS, and the washing volume was added to the neutrophil sample. The samples were then frozen and stored at −80 °C until use. Before DNA extraction, the samples were thawed and centrifuged at 10.000× *g* at 4 °C, and the excess media was removed. The DNA was extracted using the Maxwell^®^ 16 Cell DNA Purification Kit with the Maxwell 16 Instrument (Promega, Madison, WI, USA) and quantified using the QuantiFluor™ ONEdsDNA System kit (Promega, Madison, WI, USA) and Quantus™ Fluoremeter (Promega, Madison, WI, USA). The extracted DNA was stored at −80 °C until qPCR was performed. A total of 15 ng of DNA for the neutrophil samples or 5 µL of DNA for the supernatant samples was added to each qPCR reaction. This difference between the supernatants and neutrophils was established based on the low DNA concentration of the supernatant samples.

The primers used for the *Map* IS900 qPCR were forward (MP10-1, [5′-ATGCGCCACGACTTGCAGCCT-3′]) and reverse (MP11-1, [5′-GGCACGGCTCTTGTTGTAGTCG-3′]) [45]. The detection and quantification were performed as previously described [6]. A 10-fold diluted standard curve (slope: −3.616; R^2^: 0.996; efficiency: 89.04%) was constructed using the *Map* genomic DNA, obtained from 10^8^ CFUs of the *Map* K10 strain, ranging from 10^−1^ to 10^−8^ ng of the total *Map*-DNA. In addition, to obtain a similar DNA composition to that of the samples, each standard curve point was diluted in 20 ng of goat DNA extracted from a non-infected goat lymph node. The samples were analyzed in triplicate and considered as positive when the dissociation peak (Tm) was 89.1 ± 1 °C and the threshold cycles (Ct) were ≤37. The qPCR results were analyzed using 7500 Software v2.0.6 (Applied Biosystems™). The supernatant fraction (culture supernatants) and neutrophil fraction (scraped neutrophils) and the *Map*-DNA quantity per well were calculated via interpolation of their Ct values with the standard curve and adjusted based on sample dilutions.

### 2.9. RNA Extraction, Reverse Transcription, and Quantitative Real-Time PCR

A total of 2.5 × 10^6^ neutrophils were seeded in 6-well plates and incubated with *Map* 1:10 or left untreated (control). Incubation, sample collection, RNA isolation, quantification, purity assessment, retrotranscription, cDNA dilution (to 2 ng/µL), quantitative qPCR, and data analysis were performed as previously described [23]. 

The mRNA expression levels of the cytokines (IL-1β, IL-8, TNF, TGF-β) and TLR2 were determined using quantitative real-time PCR (qRT-PCR), as described elsewhere [46]. β-actin and glyceraldehyde 3-phosphate dehydrogenase (GADPH) were used as housekeeping genes. The qPCR reactions were performed in a final volume of 20 µL, using 10 µL of PowerUp™, SYBR™ Green master mix (Applied Biosystems™, Foster City, CA, USA), 0.5 µM of each primer, and 6 ng of cDNA in a 7500 Fast Real-Time PCR System (Applied Biosystems™, CA, USA), diluted in nuclease-free water. The primers used in this study were previously described (Appendix A) [6,47,48,49]. For each target gene, a seven-point standard curve was included in each batch of amplifications based on ten-fold serial dilutions starting at 1 ng/µL of PCR product. The relative quantification of the mRNA expression levels (fold change (FC) in expression) was carried out using the comparative 2^−ΔΔCt^ method [50], and the data were expressed as the mean log_2_FC in gene expression. 

### 2.10. Statistical Analysis

The normality of the data was assessed using the Shapiro–Wilk test. All the data were normally distributed. The Levene test was used to assess the homogeneity of the variances. The ANOVA assumptions were met and different two- and three-way repeated measures ANOVA tests were performed to evaluate the effects of the factors on the different neutrophil functions studied: (i) time (day 0 versus day 30), (ii) vaccination status (non-vaccinated versus vaccinated), (iii)—only in the NETosis assays—stimuli (PMA versus *Map*), (iv) NET type (anchored versus free), and (v)—only for the *Map*-DNA—DNA location (neutrophil versus supernatant fractions). When significant interactions were detected, post hoc pairwise comparisons were performed using paired *t*-tests, and the *p*-values were adjusted using the Bonferroni multiple testing correction method; *p*-values of <0.05 were considered statistically significant. All the statistical analyses were performed using R software version 4.1.3 [51], with the packages rstatix (0.7.2), ggpubr (0.6.0.999), stringr (1.5.0), dplyr (1.1.1), purrr (1.0.1), readr (2.1.4), tidyr (1.3.0), tibble (3.2.1), and tidyverse (2.0.0).

## 3. Results

### 3.1. NETosis Visualization and Quantification

The non-stimulated neutrophils (Figure 1A) were homogenously distributed throughout the slides; their nuclei exhibited a uniform shape and size, with MPO+ granules occupying most of the cytoplasm, and only a small number of nuclei were labelled for histones. In some instances, a few solitary neutrophils could be observed releasing short, linear extracellular traps (spontaneous NETosis). The neutrophils treated with 50 nM PMA (Figure 1B) were homogeneously distributed; they seemed to be slightly larger due to the increased adhesion to the coverslips and showed abundant NETosis and, occasionally, degranulation. The neutrophils exposed to *Map* (Figure 1C) were distributed heterogeneously and frequently formed aggregates surrounding the bacterial clumps. The cell shape, MPO+ staining, and NETs varied in size and shape, and the NETs were abundant, particularly over the larger *Map* clumps. After *Map* exposure, no apparent qualitative differences were observed between the non-vaccinated and vaccinated animals in any of the samples.

The results of the quantification of NETosis through immunofluorescence can be seen in Figure 2. No significant interactions between time, stimuli, or vaccination status were detected using three-way mixed ANOVA.

The results of the DNA release calculated using fluorometry are represented in Figure 3. No significant interactions between time, stimuli, or vaccination status were detected for either the anchored or the free NETs. However, a single factor effect of the stimuli was found: PMA induced a higher increase than *Map* in both the anchored and free NETs, but this difference was only significant in the anchored NETs overall (*p* < 0.0001), by day 0 (*p* < 0.01), and by day 30 (*p* < 0.05).

### 3.2. Quantification of Neutrophil Map Phagocytosis through Flow Cytometry

No significant differences between groups or sampling days were observed in the mean percentage of GFP+ neutrophils or in the mean increase in GFP fluorescence intensity of the neutrophils incubated with *Map*-GFP (Figure 4). In the vaccinated group, a mean of 2.36% ± 0.36% GFP+ neutrophils and a mean increase of 21.90% ± 2.56% in the GFP FI of the neutrophils incubated with *Map*-GFP were observed before vaccination, whereas by 30 dpv, 2.72% ± 0.6% GFP+ neutrophils and a mean increase of 19.29% ± 7.64% in the GFP FI of the neutrophils incubated with *Map*-GFP were recorded.

### 3.3. Map qPCR

The *Map* DNA was quantified in the neutrophils and supernatants; the non-infected neutrophils and their culture supernatants were found to be negative following *Map* qPCR. The results of *Map* DNA quantification in infected wells can be seen in Figure 5, on both days and in both groups, the mean *Map* DNA quantity was significantly higher in the supernatants than in the neutrophil fraction (*p* < 0.01). No significant differences were found between the non-vaccinated and vaccinated groups on any of the days. However, on day 0, the extracellular *Map*-DNA quantities were significantly higher (*p* < 0.01) than on day 30 in both groups.

### 3.4. Differential Transcript Expression of Cytokine and TLR2 Expression by qRT-PCR

The differential mRNA expressions of TNF, IL-1β, IL-8, TGF-β, and TLR2 in the neutrophils, after incubation with *Map*, are represented in Figure 6. The β-actin and GAPDH gene expressions were similar in all the RNA samples, suggesting equivalent RNA integrity and the suitability of these genes as internal controls. In both groups and on the sampling days, the incubation with *Map*-induced meaningful increases in expression (mean log_2_FC > 1) in all the transcripts studied, except for TGF-β by day 30. 

No significant changes in the differential expression levels of any of the transcripts were observed between the vaccinated and non-vaccinated animals on either day 0 or 30. However, for both groups, the differential expression of all the transcripts studied was higher on day 0 than on day 30; this difference was significant for TNF (*p* < 0.05), IL1-β (*p* < 0.05), and IL8 (*p* < 0.0001).

## 4. Discussion

Understanding the ways in which immune cells such as macrophages, dendritic cells, monocytes, lymphocytes, or neutrophils respond to *Map* during PTB infection and after PTB vaccination is crucial for the further development of new vaccines. Considering that neutrophils are among the first cells to arrive at the tissue when an infection occurs, their role during this early phase is critical for the pathogenesis of infectious diseases. In this respect, as previously stated, their involvement in the early phase of PTB infection has been demonstrated [13]. Currently, the study of neutrophil response using ex vivo models is gaining attention due to the difficulty and limitations of in vivo functional studies. The ex vivo innate response of caprine neutrophils against *Map* has recently been described [23]. However, the possibility of neutrophil response modulation by PTB vaccination remains scarcely explored in ruminants [24], and increasing this knowledge could help in the understanding of the specific mechanisms that determine the protection against PTB after vaccination. 

Thus, in this study, the possible influence of vaccination with Gudair^®^ on goat neutrophil function was studied for the first time. The vaccine was administered to a target species in accordance with its intended use (recommended age, appropriate dosage, and designated route of administration). Key neutrophil functions were quantified by 30 dpv, a timepoint used in previous experiments on the effect of vaccination on neutrophil function [18,24,32] and in caprine experimental *Map* challenge models [52,53]; this timepoint is within the period in which animals are highly susceptible to PTB infection [54]. Under these conditions, no effect of vaccination with this heat-inactivated whole-cell vaccine on any of the neutrophil functions studied was observed. Nevertheless, previous studies have demonstrated that neutrophil function can be modulated by vaccination [18,31,32,33,35,42] as early as seven days after immunization [31]. These studies observed neutrophil function modulation when using either live attenuated vaccines [18,31,32,35,42] or a mucosally administered replicating adenoviral vector [33], all of which are capable of eliciting very strong immune responses. Specifically, the studies have mainly focused on vaccines against mycobacteria, such as BCG [31,32,35] and *Map* [18], but also on non-mycobacterial intracellular pathogens, such as *Leishmania donovani* [42] and HIV [33]. Additionally, as previously stated, most of these studies were performed in experimental models that were different from the target species of the vaccines tested. In this respect, it has been demonstrated that different oral and subcutaneous vaccines against PTB can stimulate neutrophil activity, which correlates with protection in rabbits [18]. Meanwhile, recent studies in sheep [55] and goats [56] have associated the protective response induced by Gudair^®^ with the proliferation of several T cell subpopulations, including CD44+, CD25+, and γδ cells.

In the present study, the PTB vaccination of goats with Gudair^®^ did not influence the degree of NETosis induced by *Map* in vitro. This finding is in line with that observed after the vaccination of sheep with Silirum^®^ following a similar experimental design [24]. This was observed previously in other mycobacterial vaccines, which did not affect NETosis, such as an attenuated oral vaccine against *Map* in rabbits [18] and BCG in humans [32]. However, these findings contrast with an increase in NETosis against *Map* and other pathogens observed after vaccination of rabbits with Silirum^®^ [18]. In this regard, the rabbit model has proven to be a useful alternative to ruminants in the study of PTB pathogenesis and vaccination [57]. However, there is a lack of comparative studies on neutrophil (or other leukocytes) function, and host–pathogen interactions are not always extrapolatable between species, as has already been confirmed between mice and humans [58,59]. Additionally, Silirum^®^ and Gudair^®^ differ in the adjuvant employed and only share the same antigen. When comparing these studies, it is noteworthy that the same final dose of antigen (2.5 mg) was administered via the subcutaneous route to rabbits [18] and sheep [24] and to the goats in the present experiment. The higher relative dose of antigen administered to rabbits, in relation to the live weight, could be another important factor when explaining this discrepancy.

As previously mentioned, in vitro NETosis against *Map* has recently been described in rabbits and ruminants [16,18,23,24] and has been observed in vivo in a guinea pig model of *Mtb* infection [25]. The NETs visualized and quantified through microscopy-based methods would correspond to anchored NETs—those attached to the neutrophils. On the other hand, the quantification of NETs through methods based on the determination of DNA concentration generally do not discriminate between anchored and cell-free NETs. In this regard, cell-free NETs have been described in vivo and in vitro in murine models of sepsis and cancer and have been shown to trap circulating bacteria [44,60] or tumoral cells [61]. This type of NET is still poorly defined and studied, and the effects of this entrapment appear to be either positive, with circulating cell-free NETs minimizing the vascular spread of bacteria [60], or negative, as these NETs have been associated with the formation of platelet and leukocyte aggregates during sepsis [44], and they sequester tumor cells, favoring metastasis [61]. Cell-free NETosis against *Besnoitia besnoiti* has been quantified in vitro in bovines, using a similar method to that applied in this study, but its possible implications in vivo were not discussed [62]. To our knowledge, this type of NET has not previously been studied in PTB or other mycobacterial infections. Therefore, we can only hypothesize about this cell-free NETs origin or its possible in vivo implications; these NETs could simply be a result of the breakage of anchored NETs by *Map* extracellular DNAses, as recently described in *Map*-K10 [26]. In any case, if these cell-free NETs are spontaneously released by neutrophils, their implications in the control of *Map* spread to other organs in vivo would probably be limited, as *Map* dissemination mainly takes place inside migrating macrophages and dendritic cells [63]. However, they could have a role at an earlier timepoint in *Map* pathogenesis, as intestinal intraluminal NETs have recently been described in *Salmonella* gut infection [19]; the authors of the latter study hypothesized that NETs may not only immobilize bacteria but may also contribute to their removal with the fecal flow.

Regarding *Map* phagocytosis by neutrophils, in the present experiment, no significant differences between vaccinated and non-vaccinated animals were found, suggesting that, under these experimental conditions, vaccination does not modulate the phagocytic activity of neutrophils against *Map*. As with the results on NETosis, this finding diverges from those in rabbits, where an increase in *Map* phagocytosis was measured after immunization against PTB with inactivated or live attenuated vaccines [18]. There were, however, differences in the experimental designs, such as different *Map* vaccines (Gudair^®^ versus Silirum^®^ or oral vaccination) or the incubation of neutrophils and *Map* in the presence of autologous serum, which may suggest that opsonization of *Map* could play a role in this mechanism. Murine neutrophils from mice vaccinated with BCG also showed an increase in the phagocytosis of *S. aureus* bioparticles, revealing a heterologous effect, though this effect was only studied at 7 dpv [32]. In the current study, and in order to analyze the different mechanisms involved in neutrophil phagocytosis, two different methods were used: an FCM assay, which was previously used to quantify neutrophil phagocytosis of *Map* [18,23] and *Staphylococcus aureus* [64], and the quantification of *Map* DNA using qPCR, which was previously applied in a similar manner to quantify macrophage phagocytosis of *Map* in vitro [6]. Each technique provides different information on neutrophil phagocyting activity and at different timepoints, 20 min and 3 h. On the one hand, the percentage of neutrophils phagocyting *Map*, as measured through an FCM FI threshold will only represent the neutrophils that phagocyte high amounts of *Map*-GFP; this could offer an insight into the differences between distinct neutrophil subsets or activation states [65,66,67]. Additionally, the mean increase in FI is more apparent when detecting subtle changes in *Map*-GFP phagocytosis and taking into consideration neutrophils which phagocyte low numbers of bacteria. Finally, the quantification of *Map* DNA in the supernatant and neutrophil fractions allows the relative quantification of both live and dead bacteria after a long incubation time, as GFP degradation and loss of fluorescence after phagosome acidification [68] could be a relevant issue in this case. However, this method does not differentiate between phagocyted and NET-trapped bacilli, as in this assay the neutrophils are allowed to adhere and to undergo NETosis; thus, in both cases, they would be collected in the neutrophil fraction. 

It has been proposed that neutrophils participate in PTB pathogenesis precisely by starting a very early effector response against *Map* [16]. According to this hypothesis, neutrophils would phagocytose and immobilize mycobacteria with their NETs and then release IL-1β, triggering a proinflammatory response. IL-1β would induce macrophage transepithelial migration to the site of infection and activation [69]. Upon arrival, the macrophages, in an efferocytosis process, would engulf NETs and the apoptotic remnants [70], acquiring the antimicrobial compounds generated by neutrophils after the first contact with *Map*. 

Regarding this cytokine signaling, there is limited research on the possibility of modulating neutrophil cytokine production with vaccination, and it has not been studied in PTB. In monocytes, after antigen stimulation, several histone modifications result in the opening of chromatin at the promoters of genes encoding pro-inflammatory cytokines, including IL-1β and TNF, enabling a faster and stronger response upon antigen re-encounter [71]. In neutrophils, multiple expression pathways, including cytokine-specific pathways, such as those of IL-8, IL-1β, and TNF, are activated after neutrophil priming by TNF or GM-CSF [72]; this “primed” phenotype could potentially be induced by the cytokine environment established after vaccination. In fact, human BCG vaccination induces an increase in neutrophil IL-8 production in response to both homologous and heterologous stimuli (*Mtb*, *Candida albicans*, PMA, and LPS). Additionally, some epigenetic changes (part of the basis of neutrophil training), which are known to enhance gene transcription near genes encoding this cytokine and other proinflammatory cytokines such as IL-1β, were identified in vaccinated individuals [32]. Because all of this, changes in the cytokine expression profiles after vaccination should be a good indicator of the “training”, or at least the “priming”, of neutrophils. 

Nevertheless, in the present study, and in concordance with the results of the functional assays, by 30 dpv, no significant differences were found between the vaccinated and non-vaccinated animals in the neutrophil’s expression of any of the analyzed cytokines (IL-1β, IL-8, and TNF TGF-β) or TLR2 after *Map* exposure. In both groups and on the sampling days, a mild increase in TGF-β and TNF expression, accompanied by a significant increase in the expression of IL-1β and IL-8 was observed, indicating that *Map* triggers the expression of these cytokines. In fact, an increase in IL-1β and TNF expression in goat neutrophils exposed to *Map* in vitro [23] has previously been described. IL-1β is necessary for neutrophil and macrophage migration and NETosis and intervenes in the initiation of adaptive immunity, and TNF enhances the phagocytic properties of neutrophils, induces NETosis, and primes neutrophil granule exocytosis [23,69,73]. Regarding IL-8, an increase in production or expression of this cytokine has already been described in bovine neutrophils incubated with *Map* in vitro [16] and in the ileal mucosa of clinically infected animals [74]. IL-8 is constitutively expressed by neutrophils [75]; it is essential for neutrophil chemotaxis and leukocyte recruitment [76] and induces NETosis [19]. In *Mtb* infection, IL-8 plays a central role in granuloma formation [77], and it increases *Mtb* phagocytosis and killing by neutrophils and macrophages [78]. 

Neutrophils express most of the TLR family members, with the possible exception of the intracellular TLR3 and TLR7 [79], and TLR2 is probably the most important in the defense against mycobacterial pathogens [80]; its expression in circulating neutrophils is increased during *Mtb* infection [81]. In this context, neutrophil expression of TLR2 in response to *Map* has not previously been studied under our conditions; an increase in expression was induced by *Map* in both groups and on the sampling days; however, it was not affected by vaccination. TLR2 is believed to play a protective role during early infection as it triggers a strong proinflammatory response, which is considered beneficial for bacterial clearance [82]. Specifically, neutrophils are activated upon mycobacterial infection via TLR2-mediated recognition of mycobacterial lipoarabinomannan [83]; this TLR-2-mediated neutrophil activation is essential for the early induction of neutrophil IL-8 secretion [84]. The current study confirmed the role of proinflammatory cytokines and TLR2 expression in the response of neutrophils against *Map*. However, it seems that vaccination with Gudair^®^ does not modulate the expression of these molecules in goats, at least under the experimental conditions of this study. 

Whether vaccination modulates other mechanisms of the neutrophil response deserves further investigation. To summarize, our study replicated previous findings [23] regarding goat neutrophil–*Map* interactions in a larger sample size, including animals of a different age (one to two months versus eight months in the previous study) and sex. However, it is important to consider that the lack of observable differences in the vaccine’s effect on neutrophil response may be attributable to the experimental conditions employed in this study. Studies on the effects of vaccination on neutrophil effector functions are still scarce. We studied these at a single timepoint (30 dpv), which is frequently used by other works, but in some cases, this modulation was studied at very early timepoints [31,85]. It is worth noting that these results are confined to the ex vivo conditions assessed, and therefore, caution should be exercised when extrapolating them to the complex host immune response, where *Map* interacts with a diverse array of cell populations. 

## 5. Conclusions

As a main conclusion of this study, it is tempting to hypothesize that Gudair^®^’s protection against PTB in goats might not be mediated by the modulation of neutrophil phagocytosis, NETosis, or proinflammatory cytokines and TLR2 transcription. The possibility, degree, and duration of neutrophil priming or training may depend largely on the intensity of the immune response elicited by the vaccine employed or the antigen dose. Thus, it would be of interest to further study the mechanisms of protection in *Map* natural hosts, using live attenuated vaccines [18,31,32,35,42], the mucosal vaccination route [18,33], prime-boost immunization [33], or in animals of different ages or at different stages of PTB disease. 

In addition to functional assays, it would be valuable to explore novel and highly sensitive techniques for studying, under in vivo conditions, the basis of neutrophil trained immunity without the need for *Map* re-exposure, for example, histone modifications, mitogen-activated protein kinase (MAPK) phosphorylation, or long non-coding RNAs [32,34,86,87]. 

## Figures and Tables

**Figure 1 animals-14-01694-f001:**
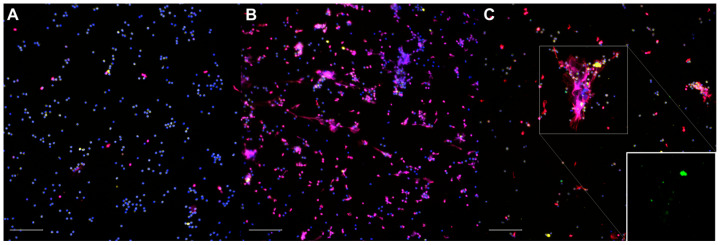
Representative micrographs of the immunofluorescence staining of neutrophils: (**A**) non-infected, unstimulated neutrophils (control); (**B**) phorbol 12-myristate 13-acetate (PMA)-stimulated neutrophils; (**C**) neutrophils exposed to *Mycobacterium avium* subspecies *paratuberculosis* (*Map*) at MOI of 10. Inset: green channel only. DNA (blue), *Map*-GFP (green), histones (red), myeloperoxidase (yellow). Micrographs were taken at 200× using the Eclipse Ni-E (Nikon, Tokyo, Japan) microscope and the Prime BSI Scientific CMOS scientific camera (Photometrics^®^ Prime BSI™, Tucson, AZ, USA). Scale bar is equal to 100 µm. Two coverslips were thoroughly assessed for each animal (*n =* 7), condition, and sampling day.

**Figure 2 animals-14-01694-f002:**
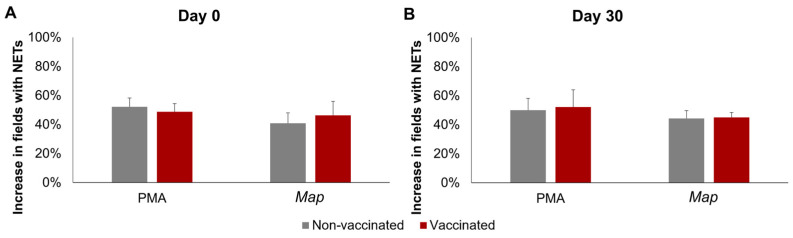
NETosis quantification through immunofluorescence. Increase in the percentage of fields (400×) with at least one neutrophil undergoing NETosis in response to phorbol 12-myristate 13-acetate (PMA) or *Mycobacterium avium* subspecies *paratuberculosis* (*Map*). (**A**) Before vaccination (day 0) and (**B**) 30 days post-vaccination (day 30). For each animal 10, 400× fields were analyzed per condition. All values for each group are means of *n* = 7, with error bars representing the standard error.

**Figure 3 animals-14-01694-f003:**
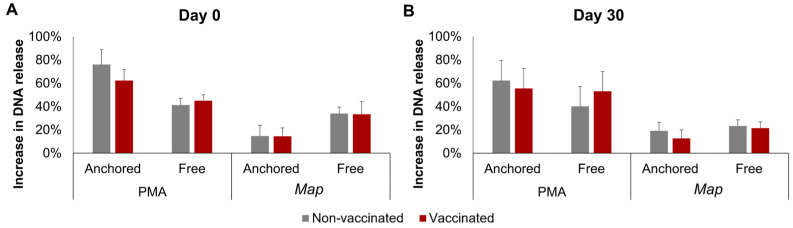
Fluorometric quantification of the increase in neutrophil DNA release. DNA release after incubation with phorbol 12-myristate 13-acetate (PMA) or *Mycobacterium avium* subspecies *paratuberculosis* (*Map*) was quantified in the supernatants and in the cells at the bottom of the plate and was considered a result of free and anchored NETosis, respectively. (**A**) Before vaccination (day 0) and (**B**) 30 days post-vaccination (day 30). For each animal, neutrophil seeding, culture, and DNA quantification were performed in three wells per condition. All values for each group are means of *n* = 7, error bars represent the standard error.

**Figure 4 animals-14-01694-f004:**
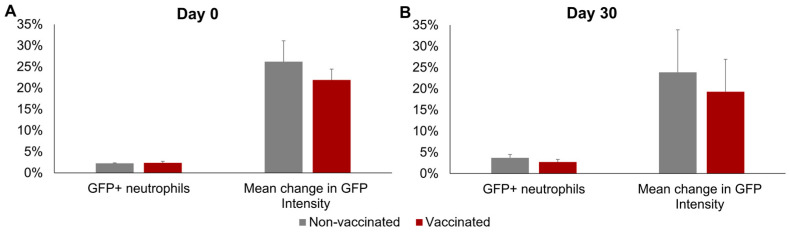
Neutrophil phagocytosis levels. Phagocytosis against *Mycobacterium avium* subspecies *paratuberculosis* (*Map*), measured as the percentage of GFP+ neutrophils and the increase in GFP intensity in the neutrophils after incubation with *Map*-GFP. (**A**) Before vaccination (day 0) and (**B**) 30 days post-vaccination (day 30). All values for each group are means of *n* = 7, with error bars representing the standard error.

**Figure 5 animals-14-01694-f005:**
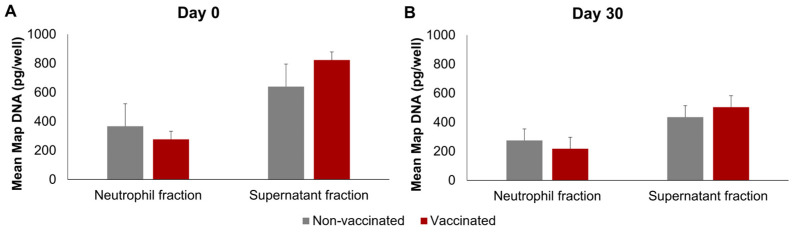
Quantification of *Map* DNA using qPCR. (**A**) Before vaccination (day 0) and (**B**) 30 days post-vaccination (day 30). qPCR was performed in triplicate for each sample. All values for each group are means of *n* = 7, with error bars representing the standard error.

**Figure 6 animals-14-01694-f006:**
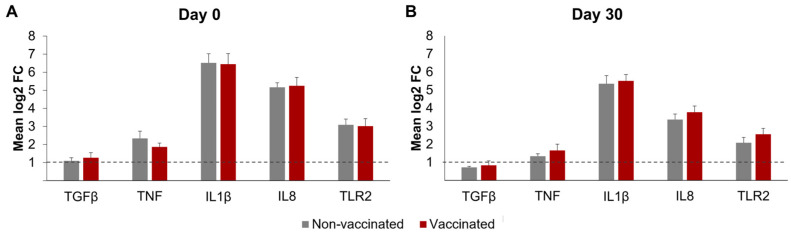
Differential transcript expression of neutrophils after incubation with *Mycobacterium avium* subspecies *paratuberculosis*. Bar plots represent the mean log_2_-fold change in gene expression, as determined by qRT-PCR of the genes TGF-β, TNF, IL-1β, IL8, and TLR2. (**A**) Before vaccination (day 0) and (**B**) 30 days post-vaccination (day 30). Dotted lines show the threshold of log_2_ fold change ≥1. qPCR was performed in duplicate for all samples and genes studied. All values for each group are means of *n =* 7. All values are means, with error bars representing the standard error.

## Data Availability

The raw data supporting the conclusions of this study will be made available by the authors on request.

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
