# Peer review of "No Evidence of Neutrophil Response Modulation in Goats after Immunization against Paratuberculosis with a Heat-Inactivated Vaccine"

_animals, 2024, doi:10.3390/ani14111694_

Round 1

Reviewer 1 Report

Comments and Suggestions for Authors

The manuscript by Criado et al. is well written with an important contribution to the understanding of the effect of a commercially available vaccine on response of caprine neutrophils against Mycobacterium avium subspecies paratuberculosis. I support it’s further processing.

Overall, the title, abstract, and introduction are well-written and effectively convey the context and importance of the study.The "Materials and Methods" basically ensure the study's reproducibility and offering a clear understanding of the techniques used. The results are well presented. The discussion section effectively interprets the results and provides a deeper understanding of their significance. The paper's novelty and the way it presents the properties under investigation make a strong case for its acceptance in its current form.

According to the journal requirement (https://www.mdpi.com/journal/animals/instructions) and before to the Abstract section, the authors must complete the manuscript with a simple summary, consisting of no more than 200 words.

The establishing of the total number of enrolled animals in the study (fourteen) is unclear. The authors must convince the reviewer that the results provided by a limited number of animals are completely supportable by statistical tools.

L553: “no differences were found” – unclear statement, please be more specific

The authors must insert a separate Conclusion section, highlighting the main findings, the study limitations and indicating future research directions in the approached topic.

L596: “We hypothesize” – please avoid the personal mode formulations within scientific papers, may sound unprofessional.

Within the reference section, the authors must paid special attention on the italics writing of the scientific name of the species. Please carefully revise this concern.

Author Response

Thank you for your review. We have carefully considered all your comments and consequently revised the manuscript. In addition to incorporating the simple summary, we have made several significant changes throughout the manuscript.

According to the journal requirement (https://www.mdpi.com/journal/animals/instructions) and before to the Abstract section, the authors must complete the manuscript with a simple summary, consisting of no more than 200 words.

The Simple summary before the Abstract has been added to the Manuscript:

“Simple summary: Among other limitations, the currently available vaccines against paratuberculosis do not offer complete protection against infection, and further vaccine development is limited by a lack of understanding of the mechanisms behind vaccine-induced protection. In this regard, the most recent studies have demonstrated that neutrophil function can be modulated through vaccination against several pathogens, including Mycobacterium avium subspecies paratuberculosis (Map). However, this modulation has not been described in ruminants, which are the natural hosts of Map. In the present work, the effect of vaccination on the neutrophil response against Map was assessed in goats using the only available vaccine against small ruminant paratuberculosis, Gudair®. No differences were found in the ex vivo response of neutrophils isolated from non-vaccinated and vaccinated animals, which suggests that the protection conferred by this heat-inactivated vaccine is based on mechanisms other than neutrophil modulation. It is possible that neutrophil modulation depends largely on the intensity of the immune response elicited by the vaccine employed or the antigen dose, as the previous reports which observed it, used live attenuated vaccines or were performed in laboratory animals using experimental vaccines.”

The establishing of the total number of enrolled animals in the study (fourteen) is unclear. The authors must convince the reviewer that the results provided by a limited number of animals are completely supportable by statistical tools.

Previous works in ruminants, which applied similar techniques for the study of neutrophil function against different pathogens, found significant effects of the treatments/stimuli using low sample sizes. For example:

  • (Silva et al., 2014) (Goat, n=3).
  • (Caro et al., 2014; Villagra-Blanco et al., 2017) (Cattle, n=3).
  • (Ladero-Auñon, Molina, Holder, et al., 2021) (Cattle, Map, n=4).
  • (Criado et al., 2023) (Goat, Map, n=3).

Studies on the effect of vaccination on neutrophil function are scarce and most were performed either in mice on in humans. A previous study on paratuberculosis vaccination, performed in rabbits used a sample size of 5 animals per group (Ladero-Auñon, Molina, Oyanguren, et al., 2021). This n was enough for demonstrating the effect of vaccination or Map challenge on neutrophil function, for example, at 1-month post-vaccination neutrophils from Silirum®-vaccinated rabbits showed a significantly (p < 0.001) higher Map phagocytosis (≈5% vs ≈20%); 3 months post-vaccination and 2 post-challenge, they showed significant (p < 0.01) higher NETosis against Map (≈2.5-fold change). There is no research on the effect that vaccination has on neutrophil cytokine expression, but it has been demonstrated that human BCG vaccination increases (≈2-fold change) neutrophil IL-8 production against M. tuberculosis and other stimuli (Moorlag et al., 2020).

Nevertheless, the in vivo biological relevance of the results from most of the mentioned ex vivo assays remain uncertain. On the other hand, mononuclear cell function is well-known to be modulated by vaccination and involved in vaccine induced-protection, and several works have demonstrated that vaccines induce very significant changes in the mononuclear cells cytokine expression following antigen re-exposure (often with fold changes well over 5), for example: goat macrophages cultured with Map (Arteche-Villasol et al., 2021), rabbit PBMCs cultured with Map  (Ladero-Auñon, Molina, Oyanguren, et al., 2021) or PPD stimulated whole-blood from BCG-vaccinated humans (Matsumiya et al., 2015).

However, the experimental design among these greatly differs, thus complicating power calculation. Post-hoc power calculation cannot be performed given the lack of differences between the groups of this study. Therefore, only an a priori power calculation can be performed, with the Cohen’s effect size roughly estimated from the previous examples. A priori sample size calculations were performed in G*power 3.1.9.7. for a repeated measures ANOVA within-between interaction with two groups, two measures, minimum power = 80% and an α = 0.05:

  1. Phagocytosis assay (Ladero-Auñon, Molina, Oyanguren, et al., 2021):

Non-vaccinated group: 5 ± 5 %; Vaccinated group:  20 ± 10 %. à f = 1.89

Total sample size= 6. Actual power= 92%

  1. NETosis assay (Ladero-Auñon, Molina, Oyanguren, et al., 2021):

Non-vaccinated group: 1 ± 0.5; Vaccinated group:  2.5± 1  à f = 1.34

Total sample size = 8. Actual power = 92%

  1. Cytokine expression A (Moorlag et al., 2020) (Neutrophil IL-8 production*):

*Gene expression does not correlate well with protein production (Vogel y Marcotte, 2012).

Non-vaccinated group: 1 ± 0.2; Vaccinated group:  1.8 ± 0.2à f =4

Total sample size = 4. Actual power = 96%

  1. Cytokine B (Matsumiya et al., 2015; Arteche-Villasol et al., 2021; Ladero-Auñon, Molina, Oyanguren, et al., 2021) (Cytokine expression based on a FC=5):

Non-vaccinated group: 1 ± 0.5; Vaccinated group:  5 ± 3à f = 1.86

Total sample size = 6. Actual power = 91%

All total sample size estimates were well below the one employed in this study (14) suggesting that our sample size should be sufficient. However, since the null hypothesis can never be proven true, we have expressed our conclusions with appropriate uncertainty statements.

Arteche-Villasol, N., Gutiérrez-Expósito, D., Vallejo, R., Espinosa, J., Elguezabal, N., Ladero-Auñon, I., Royo, M., del Carmen Ferreras, M., Benavides, J. y Pérez, V. (2021) "Early response of monocyte-derived macrophages from vaccinated and non-vaccinated goats against in vitro infection with Mycobacterium avium subsp. paratuberculosis", Veterinary Research. BioMed Central Ltd, 52(1), pp. 1-12."

Caro, T. M., Hermosilla, C., Silva, L. M. R., Cortes, H. y Taubert, A. (2014) "Neutrophil extracellular traps as innate immune reaction against the emerging apicomplexan parasite Besnoitia besnoiti", PLoS ONE. Public Library of Science, 9(3), p. 91415. doi:10.1371/JOURNAL.PONE.0091415.

Criado, M., Pérez, V., Arteche-Villasol, N., Elguezabal, N., Molina, E., Benavides, J. y Gutiérrez-Expósito, D. (2023) "Evaluation of the innate immune response of caprine neutrophils against Mycobacterium avium subspecies paratuberculosis in vitro", Veterinary research. Vet Res, 54(1), p. 61. doi:10.1186/S13567-023-01193-7.

Ladero-Auñon, I., Molina, E., Holder, A., Kolakowski, J., Harris, H., Urkitza, A., Anguita, J., Werling, D. y Elguezabal, N. (2021) "Bovine neutrophils release extracellular traps and cooperate with macrophages in Mycobacterium avium subsp. paratuberculosis clearance in vitro", Frontiers in Immunology. Frontiers Media S.A., 12, p. 645304. doi:10.3389/FIMMU.2021.645304.

Ladero-Auñon, I., Molina, E., Oyanguren, M., Barriales, D., Fuertes, M., Sevilla, I. A., Luo, L., Arrazuria, R., Buck, J. De, Anguita, J. y Elguezabal, N. (2021) "Oral vaccination stimulates neutrophil functionality and exerts protection in a Mycobacterium avium subsp. paratuberculosis infection model", NPJ vaccines. NPJ Vaccines, 6(102), pp. 1-15. doi:10.1038/S41541-021-00367-8.

Matsumiya, M., Satti, I., Chomka, A., Harris, S. A., Stockdale, L., Meyer, J., Fletcher, H. A. y McShane, H. (2015) "Gene Expression and Cytokine Profile Correlate With Mycobacterial Growth in a Human BCG Challenge Model", The Journal of Infectious Diseases. Oxford University Press, 211(9), p. 1499. doi:10.1093/INFDIS/JIU615.

Moorlag, S. J. C. F. M., Rodriguez-Rosales, Y. A., Gillard, J., Fanucchi, S., Theunissen, K., Novakovic, B., de Bont, C. M., Negishi, Y., Fok, E. T., Kalafati, L., Verginis, P., Mourits, V. P., Koeken, V. A. C. M., de Bree, L. C. J., Pruijn, G. J. M., Fenwick, C., van Crevel, R., Joosten, L. A. B., Joosten, I., Koenen, H., Mhlanga, M. M., Diavatopoulos, D. A., Chavakis, T. y Netea, M. G. (2020) "BCG Vaccination Induces Long-Term Functional Reprogramming of Human Neutrophils", Cell Reports. Cell Press, 33(7), p. 108387. doi:10.1016/J.CELREP.2020.108387.

Silva, L. M. R., Muñoz Caro, T., Gerstberger, R., Vila-Viçosa, M. J. M., Cortes, H. C. E., Hermosilla, C. y Taubert, A. (2014) "The apicomplexan parasite Eimeria arloingi induces caprine neutrophil extracellular traps", Parasitology Research. Springer Verlag, 113(8), pp. 2797-2807. doi:10.1007/S00436-014-3939-0/FIGURES/6.

Villagra-Blanco, R., Silva, L. M. R., Muñoz-Caro, T., Yang, Z., Li, J., Gärtner, U., Taubert, A., Zhang, X. y Hermosilla, C. (2017) "Bovine polymorphonuclear neutrophils cast neutrophil extracellular traps against the abortive parasite Neospora caninum", Frontiers in Immunology. Frontiers Media S.A., 8(MAY), p. 606. doi:10.3389/FIMMU.2017.00606/BIBTEX.

Vogel, C. y Marcotte, E. M. (2012) "Insights into the regulation of protein abundance from proteomic and transcriptomic analyses", Nature Publishing Group. doi:10.1038/nrg3185.

L553: “no differences were found” – unclear statement, please be more specific

This phrase has been changed to a more specific statement:

Line 573 " Nevertheless, in the present study, and in concordance with the results of the functional assays, by 30 dpv no significant differences were found between the vaccinated and non-vaccinated animals in the neutrophil’s expression of any of the analyzed cytokines (IL-1β, IL-8, and TNF TGF-β) or TLR2 after Map exposure."

The authors must insert a separate Conclusion section, highlighting the main findings, the study limitations and indicating future research directions in the approached topic.

The conclusions of the paper have been moved to a separated "Conclusions" section:

5. Conclusions

As a main conclusion of this study, it is tempting to hypothesize that Gudair®’s protection against PTB in goats might not be mediated by the modulation of neutrophil phagocytosis, NETosis, or proinflammatory cytokines and TLR2 transcription. The possibility, degree, and duration of neutrophil priming or training may depend largely on the intensity of the immune response elicited by the vaccine employed or the antigen dose. Thus, it would be of interest to further study the mechanisms of protection in Map natural hosts, using live attenuated vaccines [18,31,32,35,42], the mucosal vaccination route [18,33], or prime-boost immunization [33] or in animals of different ages or at different stages of PTB disease.

In addition to functional assays, it would be valuable to explore novel and highly sensitive techniques for studying, under in vivo conditions, the basis of neutrophil trained immunity without the need for Map re-exposure, for example, histone modifications, mitogen-activated protein kinase (MAPK) phosphorylation, or long non-coding RNAs [32,34,86,87].”

L596: “We hypothesize” – please avoid the personal mode formulations within scientific papers, may sound unprofessional.

This formulation has been removed.

Within the reference section, the authors must paid special attention on the italics writing of the scientific name of the species. Please carefully revise this concern.

All scientific names have now been italicized in the reference section.

Reviewer 2 Report

Comments and Suggestions for Authors

In this study, researchers have studied the vaccination induced NETosis over the non-vaccinated goats using commercial vaccine Guadir developed for paratuberculosis. Overall study failed to determine the statistcial differcnes between non vaccinated and vaccinated goats in ex-vivo NETosis model. 

Study needs to evidence to show Mab induces NETosis in NETosis quantification studies comparing with non-stimulated neutrohils to rule out the probability of nonspecific NETosis. Timing is also essential to show the enhancement of NETosis. If it`s possible, authors need to show time dependent floruosnce intensity changes over the time comparing unstimulated controls,  PMA induced and Mab induced neutrophils. 

I also don`t think that collecting blood after 30d of vaccination will yield such significant results unless memory response was expected from Neutrophils which is something different than in this study design showed. The correct timing should be around 3 to 7 days postvaccination to seek neutosis from the vaccinated animals.  I don`t see a relevance to show in vitro neutrophil activity on day 30 of vaccination. 

Author Response

Thank you for your review. We have carefully considered all your comments and subsequently revised the manuscript. Below are our responses to your comments and suggestions:

Study needs to evidence to show Mab induces NETosis in NETosis quantification studies comparing with non-stimulated neutrohils to rule out the probability of nonspecific NETosis. Timing is also essential to show the enhancement of NETosis. If it`s possible, authors need to show time dependent floruosnce intensity changes over the time comparing unstimulated controls,  PMA induced and Mab induced neutrophils. 

We agree with the reviewer. It is important to take into account non-specific NETosis in neutrophil ex vivo studies it is important. As indicated in the section 2.6. NETosis quantification, the results of both assays where NETosis was quantified were obtained by subtracting the non-specific NETosis, measured in the non-stimulated/control wells (neutrophils cultured in the same conditions but without Map or PMA):

For immunofluorescence (Line 240): " The data were expressed as an increase in fields with NETs and were obtained by subtracting the percentage of positive fields in the control samples from the treated samples."

For fluorometry (Line 253): "... for each animal, the increase in anchored and cell-free NETosis, induced by PMA and Map, with respect to the untreated neutrophils, was calculated using the mean DNA con-centration of each sample and expressed as a percentage.”

I also don`t think that collecting blood after 30d of vaccination will yield such significant results unless memory response was expected from Neutrophils which is something different than in this study design showed. The correct timing should be around 3 to 7 days postvaccination to seek neutosis from the vaccinated animals.  I don`t see a relevance to show in vitro neutrophil activity on day 30 of vaccination. 

We agree with the reviewer in that studying the possibility of modulation few days after vaccination could be of interest, specially in order to investigate the basic mechanism of trained immunity. However, the 30 days post vaccination time point was chosen on the basis of previous works which have studied the effect of vaccination on neutrophil function, as stated in the introduction.  Those previous reports have been made on experimental models or experimental vaccines and have reported a memory response, i.e. trained response, in that time point. One of the main objectives of this study was to determine whether a commercially available vaccine, that have shown a degree of effectiveness, could also modulate the response of neutrophils at the same time point. We have now added a justification of the sampling time choice in the discussion section:

Line 456: " Key neutrophil functions were quantified by 30 dpv, a timepoint used by previous experiments on the effect of vaccination on neutrophil function [18,24,32] and in caprine experimental Map challenge models [52,53]; this timepoint is within the period in which animals are highly susceptible to PTB infection [54]"

Reviewer 3 Report

Comments and Suggestions for Authors

Dear Authors,

the topic on involvement and modulation of neutrophils ex vivo after PTB/Map vaccination addressed by you is interesting also to understand PTB pathogenesis, and moreover the work is well-designed and well presented.

Anyway I have some doubts and suggestions.

The title is too direct, the assumption is quite "strong" also beacuse the animals enrolled in your study are partly limited (7 vaccinated and 7 not). Your observations are potential and they have to be confirmed in the future, as you also state in the final passages of the Discussion.

In the Abstract please define NET-osis.

In the keywords, I suggest to add "cell-mediated immunity".

It would have been interesting and probably "ambitious" evaluating the neutrophils envolvement and modulation not only in vaccinated or not vaccinated goat's flock without previous history of PTB, but also in relation to the type of flock: officially PTB free flock; PTB positive flock (if present); flocks with no serological positivities. This categorization together with the adoption of gamma-IFN assay, with different interpretative criteria for PTB diagnosis (even if in bovine farms) is deepened in the publication of Corneli et al., 2021 "Early Detection of Mycobacterium avium subsp. paratuberculosis Infected Cattle: use of Experimental Johnins and Innovative Interferon-Gamma Test Interpretative Criteria".

You could extend your analyses, also in the future.

In the subparagraph "3.3 Map qPCR" do you refer to vaccine/extracellular MAP?

I think I understand that goats come from flocks without previous history of PTB, so supposing "MAP-PCR negative".

Why did you chose only those target genes for transcriptomic and gene expression study? Other genes are implicated in neutrophils and lymphocytes pathways and route, for example gamma IFN that enhances neutrophils activity, with a key role in macrophages activation and recruitment, also in PTB.

Lines 502-504 in Discussion section, are a bit redundant. These findings have been already reported in the first part of the section, or not?

Lines 552 and following: please specify and contextualise the sampling times.

Lines 596: probably, another important variable that conditions the neutrophils and cells modulation is the age of vaccination so a vaccination response age-dependent. Please, justify and contextualise this relevant aspect and assumption.

Comments on the Quality of English Language

Minor editing of English language required

Author Response

Thank you for your revision, we have carefully considered all your comments and consequently modified the manuscript. Following your suggestions, we have changed the article tittle and made several modifications.  Below are our responses to your comments:

The title is too direct, the assumption is quite "strong" also beacuse the animals enrolled in your study are partly limited (7 vaccinated and 7 not). Your observations are potential and they have to be confirmed in the future, as you also state in the final passages of the Discussion.

Thank you for your suggestion regarding the title. We agree that the original title was too direct given the study limitations. Consequently, we have revised the title to a less direct one:

"No evidence of neutrophil response modulation in goats after immunization against paratuberculosis with a heat-inactivated vaccine"

In the Abstract please define NET-osis.

For clarity, we have changed "NETosis" for "neutrophil extracellular traps (NETs) release" in the abstract.

In the keywords, I suggest to add "cell-mediated immunity".

This keyword has been added.

It would have been interesting and probably "ambitious" evaluating the neutrophils envolvement and modulation not only in vaccinated or not vaccinated goat's flock without previous history of PTB, but also in relation to the type of flock: officially PTB free flock; PTB positive flock (if present); flocks with no serological positivities. This categorization together with the adoption of gamma-IFN assay, with different interpretative criteria for PTB diagnosis (even if in bovine farms) is deepened in the publication of Corneli et al., 2021 "Early Detection of Mycobacterium avium subsp. paratuberculosis Infected Cattle: use of Experimental Johnins and Innovative Interferon-Gamma Test Interpretative Criteria".

You could extend your analyses, also in the future.

This would be an interesting approach, but it would be in fact very ambitious, given the variability in the individual susceptibility and in the immune response of animals to PTB infection such experiment would require a significantly larger number of animals. Provided enough resources are available to perform it, it would be indeed one probable road of investigation for our group. However, we performed the current study as a first approach to address the modulation of trained immune response in an experimental model of small ruminant paratuberculosis. We believe this study could be of interest to the readers of the journal, especially as to the best of our knowledge, no similar studies have been performed in ruminants or other species.

In the subparagraph "3.3 Map qPCR" do you refer to vaccine/extracellular MAP?

We are uncertain whether we have understood this question correctly. In that section we refer to the quantification of the Map DNA in what we have termed:

  1. Supernatant or extracellular fraction, which would correspond to the Map DNA present in the collected ex vivo culture supernatants.
  2. Neutrophil or intracellular fraction, which would correspond to the Map DNA present in scraped neutrophils.

I think I understand that goats come from flocks without previous history of PTB, so supposing "MAP-PCR negative".

Yes, though there is no official PTB control program in Spain, the goat kids are acquired from a flock where no clinical cases of PTB were reported at least the past five years. In addition, it has undergone extensive testing including Map-PCR and culture throughout these years with negative results. All this in addition with the serology

Why did you chose only those target genes for transcriptomic and gene expression study? Other genes are implicated in neutrophils and lymphocytes pathways and route, for example gamma IFN that enhances neutrophils activity, with a key role in macrophages activation and recruitment, also in PTB.

Those genes were chosen because we had previously observed that the caprine neutrophil expression of some of those cytokines (TGF‑β, TNF and IL‑1β) is significantly increased after exposure to Map (Criado et al., 2023). Several cytokine-specific expression pathways like IL-8, IL-1β and TNF are activated after neutrophil priming (Wright., et al., 2013), and epigenetic changes after BCG vaccination have been described in the genes encoding IL‑1β and IL-8 (Moorlag et al., 2020).  In addition, IL-8 expression is mediated by TLR2, which is the most abundant TLR in neutrophils and is considered central in neutrophil response against mycobacteria (Gopalakrishnan et al., 2019; Kurt-Jones et al., 2020).  Most of this information and references are included in the discussion (Lines: 557-589). Due to this not being a transcriptomic study but an RT-qPCR gene expression analysis, we only have expression data on the selected genes.

References:

Criado, M.; Pérez, V.; Arteche-Villasol, N.; Elguezabal, N.; Molina, E.; Benavides, J.; Gutiérrez-Expósito, D. Evaluation of the innate immune response of caprine neutrophils against Mycobacterium avium subspecies paratuberculosis in vitro. Vet. Res. 2023, 54, 61, doi:10.1186/S13567-023-01193-7.

Wright, H.L.; Thomas, H.B.; Moots, R.J.; Edwards, S.W. RNA-Seq Reveals Activation of Both Common and Cytokine-Specific Pathways following Neutrophil Priming. PLoS One 2013, 8, e58598, doi:10.1371/journal.pone.0058598.

Moorlag, S.J.C.F.M.; Rodriguez-Rosales, Y.A.; Gillard, J.; Fanucchi, S.; Theunissen, K.; Novakovic, B.; de Bont, C.M.; Negishi, Y.; Fok, E.T.; Kalafati, L.; et al. BCG Vaccination Induces Long-Term Functional Reprogramming of Human Neutrophils. Cell Rep. 2020, 33, 108387, doi:10.1016/J.CELREP.2020.108387.

Kurt-Jones, E.A.; Mandell, L.; Whitney, C.; Padgett, A.; Gosselin, K.; Newburger, P.E.; Finberg, R.W. Role of Toll-like receptor 2 (TLR2) in neutrophil activation: GM-CSF enhances TLR2 expression and TLR2-mediated interleukin 8 responses in neutrophils. Blood 2002, 100, 1860-1868, doi:10.1182/blood.V100.5.1860.h81702001860_1860_1868.

Gopalakrishnan, A.; Dietzold, J.; Verma, S.; Bhagavathula, M.; Salgame, P. Toll-like Receptor 2 Prevents Neutrophil-Driven Immunopathology during Infection with Mycobacterium tuberculosis by Curtailing CXCL5 Production. Infect. Immun. 2019, 87, doi:10.1128/IAI.00760-18.

Lines 502-504 in Discussion section, are a bit redundant. These findings have been already reported in the first part of the section, or not?

To improve clarity and reduce redundancy we have made, some changes, the first phrase discuss the results on NETosis, and the second one on phagocytosis:

Line 481: “However, these findings contrast with an increase in NETosis against Map and other pathogens observed after vaccination of rabbits with Silirum® [18]”. 

Line 522: to " As with the results on NETosis, this finding diverges from those in rabbits, where an increase in Map phagocytosis was measured after immunization against PTB with inactivated or live attenuated vaccines [18]."

Lines 552 and following: please specify and contextualise the sampling times.

The sampling time has now been specified.

Lines 596: probably, another important variable that conditions the neutrophils and cells modulation is the age of vaccination so a vaccination response age-dependent. Please, justify and contextualise this relevant aspect and assumption.

The age choice was justified in line 151: " The recommended age for vaccination is between 1 and 6 months, and earlier timepoints seem to confer better protection [43]”.

Animal age is a well-known factor influencing both vaccination effectiveness and susceptibility to PTB infection, so this has been added in line 623: “Thus, it would be of interest to further study the mechanisms of protection in Map natural hosts, using live attenuated vaccines [18,31,32,35,42], the mucosal vaccination route [18,33], or prime-boost immunization [33] or in animals of different ages or at different stages of PTB disease.”

Reviewer 4 Report

Comments and Suggestions for Authors

The abstract does not maintain a clear focus, making it difficult to understand the main findings. The initial sentences introduce broad concepts without directly tying them to the specific study at hand.

The rationale for studying neutrophil response in the context of Gudair® vaccination is weak. The abstract fails to adequately explain why this specific aspect of the immune response was chosen for investigation, particularly given the acknowledged limited effectiveness of existing vaccines.

The abstract does not highlight any novel insights or contributions to the field. It confirms that Gudair® does not modulate neutrophil function, which could be seen as a redundant finding given existing knowledge about the vaccine's limitations.

There is insufficient detail on the methodology in abstract, such as how neutrophil functions were measured and the statistical significance of the findings.

The statement "These results suggest that protection conferred by Gudair® vaccination is based on mechanisms different to neutrophil modulation" unsupported by the data presented.

The abstract lacks a logical flow, jumping from one point to another without clear transitions. The reader is left to piece together the connections between neutrophil function, vaccine efficacy, and broader implications.

The abstract does not address potential next steps or future research directions, leaving the study feeling incomplete and unimpactful.

In the methodology section, there is no mention of control measures to account for potential confounding variables, such as environmental conditions, feed, or handling, which could affect the study’s outcomes.

The sample size of fourteen animals seems arbitrarily chosen without any justification provided. There is no discussion on whether this sample size is statistically sufficient to detect significant differences between groups.

The rationale behind choosing only female goat kids is not explained. This limitation in sample diversity could affect the generalizability of the results.

The timeline for vaccination and subsequent sampling should be clarified in a picture.

There are several grammatical errors. Proofreading and editing for language accuracy are necessary.

Comments on the Quality of English Language

The abstract does not maintain a clear focus, making it difficult to understand the main findings. The initial sentences introduce broad concepts without directly tying them to the specific study at hand.

The rationale for studying neutrophil response in the context of Gudair® vaccination is weak. The abstract fails to adequately explain why this specific aspect of the immune response was chosen for investigation, particularly given the acknowledged limited effectiveness of existing vaccines.

The abstract does not highlight any novel insights or contributions to the field. It confirms that Gudair® does not modulate neutrophil function, which could be seen as a redundant finding given existing knowledge about the vaccine's limitations.

There is insufficient detail on the methodology in abstract, such as how neutrophil functions were measured and the statistical significance of the findings.

The statement "These results suggest that protection conferred by Gudair® vaccination is based on mechanisms different to neutrophil modulation" unsupported by the data presented.

The abstract lacks a logical flow, jumping from one point to another without clear transitions. The reader is left to piece together the connections between neutrophil function, vaccine efficacy, and broader implications.

The abstract does not address potential next steps or future research directions, leaving the study feeling incomplete and unimpactful.

In the methodology section, there is no mention of control measures to account for potential confounding variables, such as environmental conditions, feed, or handling, which could affect the study’s outcomes.

The sample size of fourteen animals seems arbitrarily chosen without any justification provided. There is no discussion on whether this sample size is statistically sufficient to detect significant differences between groups.

The rationale behind choosing only female goat kids is not explained. This limitation in sample diversity could affect the generalizability of the results.

The timeline for vaccination and subsequent sampling should be clarified in a picture.

There are several grammatical errors. Proofreading and editing for language accuracy are necessary.

Author Response

We would like to acknowledge the review as we consider that the manuscript has notably improved. Among major other changes, the abstract has been modified extensively and the manuscript has undergone profesional language editing. Here are our responses to your comments:

The abstract does not maintain a clear focus, making it difficult to understand the main findings. The initial sentences introduce broad concepts without directly tying them to the specific study at hand.

The rationale for studying neutrophil response in the context of Gudair® vaccination is weak. The abstract fails to adequately explain why this specific aspect of the immune response was chosen for investigation, particularly given the acknowledged limited effectiveness of existing vaccines.

The abstract does not highlight any novel insights or contributions to the field. It confirms that Gudair® does not modulate neutrophil function, which could be seen as a redundant finding given existing knowledge about the vaccine's limitations.

There is insufficient detail on the methodology in abstract, such as how neutrophil functions were measured and the statistical significance of the findings.The statement "These results suggest that protection conferred by Gudair® vaccination is based on mechanisms different to neutrophil modulation" unsupported by the data presented.The abstract lacks a logical flow, jumping from one point to another without clear transitions. The reader is left to piece together the connections between neutrophil function, vaccine efficacy, and broader implications.The abstract does not address potential next steps or future research directions, leaving the study feeling incomplete and unimpactful.

We agree with most of the suggestions regarding the abstract, so have completely rewritten following most of your suggestions, given the constrained word limit -200 words-, and tried to highlight the importance of studying the mechanisms behind the protection that these vaccines confer, which although limited, it is enough to justify its generalized use in countries with substantial ruminant industries, such as those in the Mediterranean region and Australia. We have also tried to include some of the information in the simple summary now added.

Simple Summary: Among other limitations, the currently available vaccines against paratuberculosis do not offer complete protection against infection, and further vaccine development is limited by a lack of understanding of the mechanisms behind vaccine-induced protection. In this regard, the most recent studies have demonstrated that neutrophil function can be modulated through vaccination against several pathogens, including Mycobacterium avium subspecies paratuberculosis (Map). However, this modulation has not been described in ruminants, which are the natural hosts of Map. In the present work, the effect of vaccination on the neutrophil response against Map was assessed in goats using the only available vaccine against small ruminant paratuberculosis, Gudair®. No differences were found in the ex vivo response of neutrophils isolated from non-vaccinated and vaccinated animals, which suggests that the protection conferred by this heat-inactivated vaccine is based on mechanisms other than neutrophil modulation. It is possible that neutrophil modulation depends largely on the intensity of the immune response elicited by the vaccine employed or the antigen dose, as the previous reports which observed it, used live attenuated vaccines or were performed in laboratory animals using experimental vaccines.

Abstract: Neutrophils are believed to play a role in the initial stages of paratuberculosis, and it has recently been demonstrated that vaccination can modulate their function via priming or through epigenetic and metabolic reprogramming (training). Modulation of the neutrophil response against Mycobacterium avium subspecies paratuberculosis (Map) through vaccination has been demonstrated in a rabbit model but not in ruminants. Therefore, in the present work, the effect of vaccination on the response of caprine neutrophils against Map was studied. Neutrophils were isolated from non-vaccinated (n = 7) and Gudair®-vaccinated goat kids (n = 7), before vaccination and 30 days post-vaccination. Then, several neutrophil functions were quantified ex vivo: cell-free and anchored neutrophil extracellular trap (NET) release, phagocytosis, and the differential expression of several cytokines and TLR2. The induction of cell-free NETosis and TLR2 expression by Map is reported for the first time. However, vaccination showed no significant effect on any of the functions studied. This suggests that the protection conferred by Gudair® vaccination is based on mechanisms that are independent of the neutrophil function modulation. Further research into the impact of alternative vaccination strategies or the paratuberculosis infection stage on ruminant neutrophil function could provide valuable insights into its role in paratuberculosis.”

In the methodology section, there is no mention of control measures to account for potential confounding variables, such as environmental conditions, feed, or handling, which could affect the study’s outcomes.

To ensure consistency and reliability in our study, all experimental animals were kept under identical conditions. These conditions have been detailed comprehensively in line 144: "The goat kids were housed together in the experimental facilities of the Instituto de Ganadería de Montaña (IGM, CSIC-ULE), and followed a diet based on grass hay ad libitum and a conventional compound feed throughout the entire experiment."

The sample size of fourteen animals seems arbitrarily chosen without any justification provided. There is no discussion on whether this sample size is statistically sufficient to detect significant differences between groups.

Previous works in ruminants, which applied similar techniques for the study of neutrophil function against different pathogens, found significant effects of the treatments/stimuli using low sample sizes. For example:

  • (Silva et al., 2014) (Goat, n=3).
  • (Caro et al., 2014; Villagra-Blanco et al., 2017) (Cattle, n=3).
  • (Ladero-Auñon, Molina, Holder, et al., 2021) (Cattle, Map, n=4).
  • (Criado et al., 2023) (Goat, Map, n=3).

Studies on the effect of vaccination on neutrophil function are scarce and most were performed either in mice on in humans. A previous study on paratuberculosis vaccination, performed in rabbits used a sample size of 5 animals per group (Ladero-Auñon, Molina, Oyanguren, et al., 2021). This n was enough for demonstrating the effect of vaccination or Map challenge on neutrophil function, for example, at 1-month post-vaccination neutrophils from Silirum®-vaccinated rabbits showed a significantly (p < 0.001) higher Map phagocytosis (≈5% vs ≈20%); 3 months post-vaccination and 2 post-challenge, they showed significant (p < 0.01) higher NETosis against Map (≈2.5-fold change). There is no research on the effect that vaccination has on neutrophil cytokine expression, but it has been demonstrated that human BCG vaccination increases (≈2-fold change) neutrophil IL-8 production against M. tuberculosis and other stimuli (Moorlag et al., 2020).

Nevertheless, the in vivo biological relevance of the results from most of the mentioned ex vivo assays remain uncertain. On the other hand, mononuclear cell function is well-known to be modulated by vaccination and involved in vaccine induced-protection, and several works have demonstrated that vaccines induce very significant changes in the mononuclear cells cytokine expression following antigen re-exposure (often with fold changes well over 5), for example: goat macrophages cultured with Map (Arteche-Villasol et al., 2021), rabbit PBMCs cultured with Map  (Ladero-Auñon, Molina, Oyanguren, et al., 2021) or PPD stimulated whole-blood from BCG-vaccinated humans (Matsumiya et al., 2015).

However, the experimental design among these greatly differs, thus complicating power calculation. Post-hoc power calculation cannot be performed given the lack of differences between the groups of this study. Therefore, only an a priori power calculation can be performed, with the Cohen’s effect size roughly estimated from the previous examples. A priori sample size calculations were performed in G*power 3.1.9.7. for a repeated measures ANOVA within-between interaction with two groups, two measures, minimum power = 80% and an α = 0.05:

  1. Phagocytosis assay (Ladero-Auñon, Molina, Oyanguren, et al., 2021):

Non-vaccinated group: 5 ± 5 %; Vaccinated group:  20 ± 10 %. à f = 1.89

Total sample size= 6. Actual power= 92%

  1. NETosis assay (Ladero-Auñon, Molina, Oyanguren, et al., 2021):

Non-vaccinated group: 1 ± 0.5; Vaccinated group:  2.5± 1  à f = 1.34

Total sample size = 8. Actual power = 92%

  1. Cytokine expression A (Moorlag et al., 2020) (Neutrophil IL-8 production*):

*Gene expression does not correlate well with protein production (Vogel y Marcotte, 2012).

Non-vaccinated group: 1 ± 0.2; Vaccinated group:  1.8 ± 0.2à f =4

Total sample size = 4. Actual power = 96%

  1. Cytokine B (Matsumiya et al., 2015; Arteche-Villasol et al., 2021; Ladero-Auñon, Molina, Oyanguren, et al., 2021) (Cytokine expression based on a FC=5):

Non-vaccinated group: 1 ± 0.5; Vaccinated group:  5 ± 3à f = 1.86

Total sample size = 6. Actual power = 91%

All total sample size estimates were well below the one employed in this study (14) suggesting that our sample size should be sufficient. However, since the null hypothesis can never be proven true, we have expressed our conclusions with appropriate uncertainty statements.

Arteche-Villasol, N., Gutiérrez-Expósito, D., Vallejo, R., Espinosa, J., Elguezabal, N., Ladero-Auñon, I., Royo, M., del Carmen Ferreras, M., Benavides, J. y Pérez, V. (2021) "Early response of monocyte-derived macrophages from vaccinated and non-vaccinated goats against in vitro infection with Mycobacterium avium subsp. paratuberculosis", Veterinary Research. BioMed Central Ltd, 52(1), pp. 1-12. Disponible en: https://veterinaryresearch.biomedcentral.com/articles/10.1186/s13567-021-00940-y (Accedido: 7 de julio de 2022).

Caro, T. M., Hermosilla, C., Silva, L. M. R., Cortes, H. y Taubert, A. (2014) "Neutrophil extracellular traps as innate immune reaction against the emerging apicomplexan parasite Besnoitia besnoiti", PLoS ONE. Public Library of Science, 9(3), p. 91415. doi:10.1371/JOURNAL.PONE.0091415.

Criado, M., Pérez, V., Arteche-Villasol, N., Elguezabal, N., Molina, E., Benavides, J. y Gutiérrez-Expósito, D. (2023) "Evaluation of the innate immune response of caprine neutrophils against Mycobacterium avium subspecies paratuberculosis in vitro", Veterinary research. Vet Res, 54(1), p. 61. doi:10.1186/S13567-023-01193-7.

Ladero-Auñon, I., Molina, E., Holder, A., Kolakowski, J., Harris, H., Urkitza, A., Anguita, J., Werling, D. y Elguezabal, N. (2021) "Bovine neutrophils release extracellular traps and cooperate with macrophages in Mycobacterium avium subsp. paratuberculosis clearance in vitro", Frontiers in Immunology. Frontiers Media S.A., 12, p. 645304. doi:10.3389/FIMMU.2021.645304.

Ladero-Auñon, I., Molina, E., Oyanguren, M., Barriales, D., Fuertes, M., Sevilla, I. A., Luo, L., Arrazuria, R., Buck, J. De, Anguita, J. y Elguezabal, N. (2021) "Oral vaccination stimulates neutrophil functionality and exerts protection in a Mycobacterium avium subsp. paratuberculosis infection model", NPJ vaccines. NPJ Vaccines, 6(102), pp. 1-15. doi:10.1038/S41541-021-00367-8.

Matsumiya, M., Satti, I., Chomka, A., Harris, S. A., Stockdale, L., Meyer, J., Fletcher, H. A. y McShane, H. (2015) "Gene Expression and Cytokine Profile Correlate With Mycobacterial Growth in a Human BCG Challenge Model", The Journal of Infectious Diseases. Oxford University Press, 211(9), p. 1499. doi:10.1093/INFDIS/JIU615.

Moorlag, S. J. C. F. M., Rodriguez-Rosales, Y. A., Gillard, J., Fanucchi, S., Theunissen, K., Novakovic, B., de Bont, C. M., Negishi, Y., Fok, E. T., Kalafati, L., Verginis, P., Mourits, V. P., Koeken, V. A. C. M., de Bree, L. C. J., Pruijn, G. J. M., Fenwick, C., van Crevel, R., Joosten, L. A. B., Joosten, I., Koenen, H., Mhlanga, M. M., Diavatopoulos, D. A., Chavakis, T. y Netea, M. G. (2020) "BCG Vaccination Induces Long-Term Functional Reprogramming of Human Neutrophils", Cell Reports. Cell Press, 33(7), p. 108387. doi:10.1016/J.CELREP.2020.108387.

Silva, L. M. R., Muñoz Caro, T., Gerstberger, R., Vila-Viçosa, M. J. M., Cortes, H. C. E., Hermosilla, C. y Taubert, A. (2014) "The apicomplexan parasite Eimeria arloingi induces caprine neutrophil extracellular traps", Parasitology Research. Springer Verlag, 113(8), pp. 2797-2807. doi:10.1007/S00436-014-3939-0/FIGURES/6.

Villagra-Blanco, R., Silva, L. M. R., Muñoz-Caro, T., Yang, Z., Li, J., Gärtner, U., Taubert, A., Zhang, X. y Hermosilla, C. (2017) "Bovine polymorphonuclear neutrophils cast neutrophil extracellular traps against the abortive parasite Neospora caninum", Frontiers in Immunology. Frontiers Media S.A., 8(MAY), p. 606. doi:10.3389/FIMMU.2017.00606/BIBTEX.

Vogel, C. y Marcotte, E. M. (2012) "Insights into the regulation of protein abundance from proteomic and transcriptomic analyses", Nature Publishing Group. doi:10.1038/nrg3185.

The rationale behind choosing only female goat kids is not explained. This limitation in sample diversity could affect the generalizability of the results.

Dairy goat herds, as happens in bovine and ovine dairy industry, are mainly formed by female animals, therefore we believe that female goats are an adequate experimental model to mimic the conditions found in the field. In addition. we have obtained similar qualitative results in previous experiments regardless of the age and sex (goat neutrophils release NETs, express pro-inflammatory cytokines in response to Map, phagocyte this bacterium, etc.) (Criado et al., 2023). In this study, the use of animals of the same age and sex, from the same kidding, aids to reduce the inherent individual variability always present in outbreed individuals. Nevertheless, even though sex could certainly have some influence on the quantitative results, we believe that it is unlikely that the lack of modulation of the studied neutrophil functions that we have observed in this experiment could be affected by the sex of the animal.

The timeline for vaccination and subsequent sampling should be clarified in a picture.

Minor changes to the wording have been performed for clarity:

Line 155:" All the animals were sampled 1 day before vaccination (day 0) and 30 dpv (day 30). In each sample, blood was collected from the jugular vein into heparinized Vacutainer® tubes, to isolate the neutrophils and perform multiple ex vivo assays, as explained below. "

We believe that the vaccination and sampling schedule is sufficiently straightforward now to not require a visual representation.

There are several grammatical errors. Proofreading and editing for language accuracy are necessary.

The article has been thoroughly proofread and has undergone professional language editing. Please find the certificate attached to this response.

Round 2

Reviewer 2 Report

Comments and Suggestions for Authors

Thanks for addressing comments. 

Reviewer 3 Report

Comments and Suggestions for Authors

Dear Authors,

you have modified the manuscript accordingly, following the suggestions and taking into account the comments, in order to improve the manuscript quality and to clarify some passages in the text.

Kind regards.

Comments on the Quality of English Language

Moderate editing of English language in some passages of the text is required.

Reviewer 4 Report

Comments and Suggestions for Authors

I confirm that the manuscript is in a finalized form for publication, and there are no comments to be addressed by the authors in a revised version.